# *In vivo* genetic dissection of tumor growth and the Warburg effect

Cheng-Wei Wang[1], Arunima Purkayastha[1], Kevin T Jones[1], Shivani K Thaker[1], Utpal Banerjee[1,2,3,4]*

[1]Department of Molecular, Cell and Developmental Biology, University of California, Los Angeles, Los Angeles, United States; [2]Department of Biological Chemistry, University of California, Los Angeles, Los Angeles, United States; [3]Molecular Biology Institute, University of California, Los Angeles, Los Angeles, United States; [4]Broad Stem Cell Research Center, University of California, Los Angeles, Los Angeles, United States

**Abstract** A well-characterized metabolic landmark for aggressive cancers is the reprogramming from oxidative phosphorylation to aerobic glycolysis, referred to as the Warburg effect. Models mimicking this process are often incomplete due to genetic complexities of tumors and cell lines containing unmapped collaborating mutations. In order to establish a system where individual components of oncogenic signals and metabolic pathways can be readily elucidated, we induced a glycolytic tumor in the *Drosophila* wing imaginal disc by activating the oncogene PDGF/VEGF-receptor (Pvr). This causes activation of multiple oncogenic pathways including Ras, PI3K/Akt, Raf/ERK, Src and JNK. Together this network of genes stabilizes Hif$\alpha$ (Sima) that in turn, transcriptionally up-regulates many genes encoding glycolytic enzymes. Collectively, this network of genes also causes inhibition of pyruvate dehydrogenase (PDH) activity resulting in diminished oxphos levels. The high ROS produced during this process functions as a feedback signal to consolidate this metabolic reprogramming.

*For correspondence: banerjee@mbi.ucla.edu

## Introduction

During cancer formation, developing tumor cells acquire multiple biological capabilities that ultimately lead to malignancy. These events include, sustained proliferation, resistance to cell death, induction of angiogenesis, cellular metastasis and a reprogrammed energy metabolism (*Hanahan and Weinberg, 2011*; *Pavlova and Thompson, 2016*). Warburg discovered that many cancer cells reprogram their glucose metabolism by transitioning from oxidative phosphorylation to glycolysis even in the presence of oxygen (*Warburg, 1956a*, *1956b*; *Vander Heiden et al., 2009*). Such a metabolic state was termed 'aerobic glycolysis' and the ability of cancer cells to acquire this new metabolic state has since been referred to as the 'Warburg effect'. The phenomenon was understudied for decades, as it became clear, that contrary to Warburg's assertions, cancers were largely attributable to oncogenes and tumor suppressors, rather than to exclusive changes in metabolic status (*Huebner and Todaro, 1969*; *Stehelin et al., 1976*; *Martin, 2001*; *Knudson, 1971*; *Shih and Weinberg, 1982*). However, more recent studies have explored the link between metabolic processes and oncogenesis, and have noted that altered metabolism is an important element that contributes to the etiology of cancer (*Pavlova and Thompson, 2016*). Drugs targeting key regulators of aerobic glycolysis are being developed to be included in the cancer therapy regimen (*Weinberg and Chandel, 2015*; *Galluzzi et al., 2013*). Several oncogenic pathways, including PI3K/TOR, JNK, Ras/ERK, regulate the catalytic activity or expression of key metabolic enzymes (*Chambers and LoGrasso, 2011*; *Jones and Thompson, 2009*; *DeBerardinis et al., 2008*).

Perhaps not any longer universally supported by modern evidence in cancer-metabolism, Warburg had also proposed that cancer cells undergo a glycolytic shift for the purpose of generating the bioenergetic makeup of the rapidly dividing cell (*Warburg, 1956a*). Pyruvate is the key metabolite that is used to control the last step of glycolysis in a tumor, and in the presence of lactate dehydrogenase (LDH), pyruvate is converted to lactate. In contrast, oxidative phosphorylation requires the mitochondrial enzyme complex pyruvate dehydrogenase (PDH) that converts pyruvate to acetyl-CoA, essential for the initiation of the tricarboxylic acid (TCA) cycle (*Linn et al., 1969*; *Leiter et al., 1978*). PDH is rendered inactive when it is phosphorylated by pyruvate dehydrogenase kinase (PDHK) and is activated when dephosphorylated by the phosphatase, PDHP (*Harris et al., 2002*; *Bowker-Kinley et al., 1998*). Many cancers maintain high ox-phos as well as glycolysis, maximizing the anapleuretic functions of the cell that provide the building blocks for lipid, protein and nucleotide synthesis.

The mammalian transcription factor, Hypoxia-inducible factor-1α (Hif-1α; called Sima or Hifα in *Drosophila*), regulates a number of target genes that promote various aspects of cancer, including metabolism, angiogenesis, cell survival, drug resistance, and invasive motility (*Wykoff et al., 2000*; *Carmeliet et al., 1998*; *Ryan et al., 1998*; *Pennacchietti et al., 2003*; *Ema et al., 1997*; *Semenza, 2003*; *Dang and Semenza, 1999*). Hif-1α participates in this process as hypoxia favors glycolysis over oxidative phosphorylation for ATP generation (*Zhong et al., 2000*; *Keith and Simon, 2007*; *Bertout et al., 2008*; *Dang and Semenza, 1999*; *Kim et al., 2006*). Hypoxia has been the proposed mechanism for oncogenes to effect a change in metabolic state (*Finley et al., 2013*; *Ying et al., 2012*; *Vander Heiden et al., 2009*; *Levine and Puzio-Kuter, 2010*; *Fukuda et al., 2002*). Mammalian studies often involve immortalized cell-lines with a variable and often unknown genetic background. Furthermore, while initiation of glycolysis has been studied (*Lunt and Vander Heiden, 2011*), the mechanism for the maintenance of the altered metabolic state under normoxic conditions is not as clear. Using *Drosophila* as a model system, we provide here, a complete genetic dissection of one mechanism that leads to and sustains a metabolic reprogramming in which Hifα, but not hypoxia, plays an important role.

Hif-1α and c-Jun N-terminal kinase (JNK) are associated together in many tumor types (*Comerford et al., 2004*; *Laderoute et al., 2004*; *An et al., 2013*). It is well established that reactive oxygen species (ROS) such as superoxide and peroxide radicals can cause both activation of the JNK pathway (*Lo et al., 1996*; *Owusu-Ansah and Banerjee, 2009*) and stabilization of Hif-1α (*Dröge, 2002*; *Chandel et al., 2000*). It is increasingly apparent that persistent activation of JNK signaling is involved in cancer development, progression and perhaps cellular transformation (*Manning and Davis, 2003*; *Raitano et al., 1995*; *Smeal et al., 1991*; *Wagner and Nebreda, 2009*). In addition to the above functions, it is likely that JNK could have an indirect role in attenuating oxidative phosphorylation by activating PDHK, thus blocking PDH function (*Zhou et al., 2009, 2008*). Determining how a variety of oncogenic pathways interact together to cause the metabolic reprogramming from oxidative phosphorylation to glycolysis is the central focus of this investigation. We achieve this by activating a single oncogene and show that this leads to a cascade of events that ultimately cause a glycolytic activation and allow maintenance of this altered metabolic state. There are multiple ways to model the 'Warburg effect'. This study takes advantage of the powerful genetic techniques in *Drosophila* used to identify epistatic relationships to provide a comprehensive and mechanistic basis for the establishment and maintenance of this metabolic transition in a receptor tyrosine kinase (RTK) induced tumor.

## Results

### LDH activation and transcription by a specific RTK

Aerobic glycolysis in tumors is characterized by the conversion of pyruvate to lactate by the enzyme, lactate dehydrogenase (LDH). Importantly, LDH has been demonstrated to be a marker for poor prognosis in multiple malignancies such as renal cell carcinoma (*Armstrong et al., 2012*). The *Drosophila* genome contains a single gene encoding an LDH enzyme (*ImpL3*), and biochemical studies demonstrate that it functions most like LDHA, the human form predominantly expressed in skeletal muscle that favors the conversion of pyruvate to lactate (*Rechsteiner, 1970*). An increase in LDHA enzymatic activity has been observed in diverse malignant cancers (*Dang and Semenza, 1999*).

We adapted a classic biochemical enzymatic assay (*Abu-Shumays and Fristrom, 1997*) to visualize the activity of LDH in vivo. Endogenous LDH activity is not observed in epithelial imaginal tissues such as the wing and eye discs (*Figure 1a*; *Figure 1—figure supplement 1a*) that developmentally mature in the larva to give rise to the corresponding appendages in the adult (*Swammerdam, 1737*), although evidence for such endogenous expression can be observed in the brain and salivary glands (*Figure 1—figure supplement 1c–d*). Using the Gal4/UAS system (*Brand and Perrimon, 1993*) to individually activate oncogenes in the wing disc, we found that the activated PDGF/VEGF receptor (known as Pvr) causes robust LDH activity (*Figure 1b*) while many other oncogenes do not (see later).

To determine whether the increase in LDH activity is due to an increase in *LDH* transcription, a GFP-based enhancer trap (*Quiñones-Coello et al., 2007*) was used to visualize *LDH* expression. This reporter (called *LDH-GFP*) consists of *EGFP* inserted within 50 base pairs upstream of the LDH transcriptional start site within its native locus. *LDH-GFP* is a direct insertion of GFP into the endogenous *LDH* locus and is not affected when combined with *UAS/Gal4* constructs. As expected, no GFP is detected in the developing wild-type wing or eye disc of *LDH-GFP* larvae that are otherwise wild type (*Figure 2c*; *Figure 1—figure supplement 1b*), and similar to the results of the activity assay, endogenous *LDH-GFP* expression is readily apparent in the brain and in the salivary gland (*Figure 1—figure supplement 1c–d*). However, in a *dpp*$^{blk1}$*Gal4, UAS-Pvr*$^{act}$genetic background, in which Pvr$^{act}$ is mis-expressed along the anterior/posterior boundary of the wing disc, a large tumorous overgrowth is observed with the mutant cells exhibiting robust *LDH-GFP* expression (*Figure 1d*). Ras1$^{act}$, functioning downstream, also causes some *LDH* expression, but this effect is

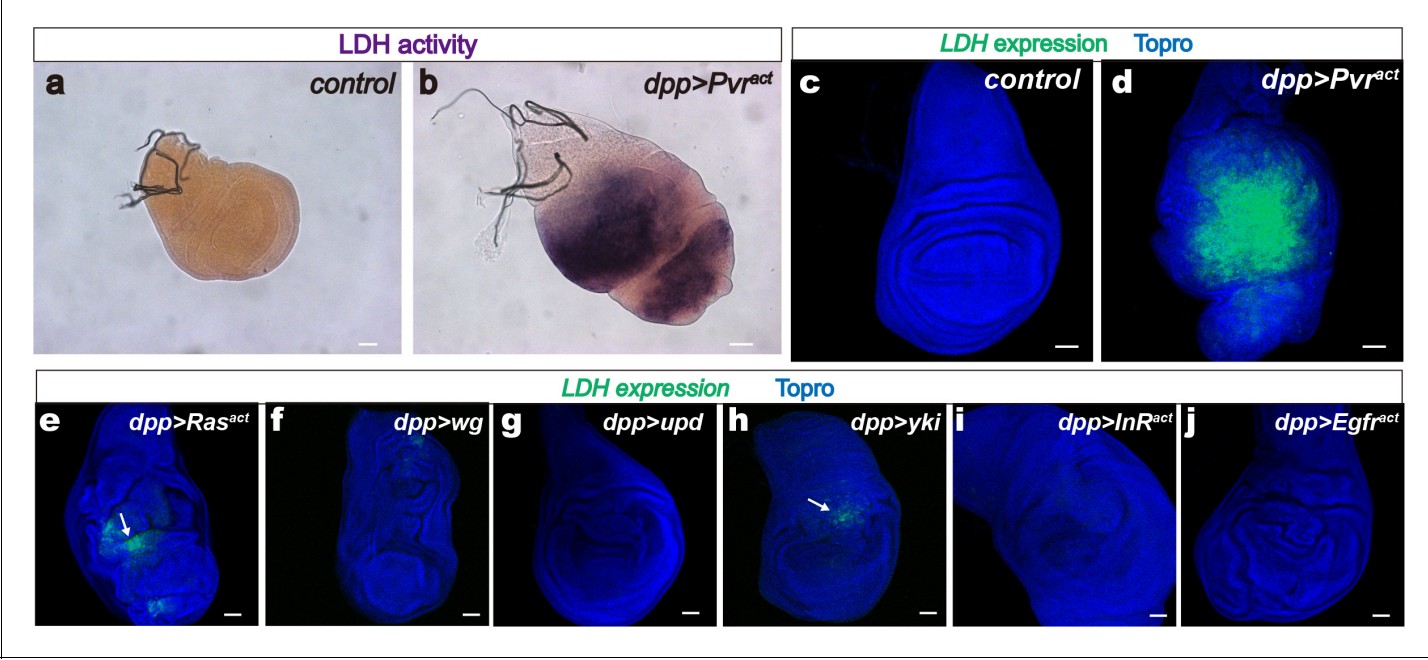

**Figure 1.** *LDH* induction by a single activated oncogene. Wing discs are from wandering third instar larvae. Scale bars, 50 μm. (a–b) LDH enzymatic activity is induced by Pvr$^{act}$ (full genotype: *dpp*$^{blk-1}$*-Gal4, UAS- GFP, UAS-Pvr*$^{act}$) stained for LDH activity (brown precipitate). The green channel is not shown in this panel for clarity. (a) Control, (*dpp-Gal4, UAS-GFP*) no detectable LDH enzymatic activity. (b) Expression of Pvr$^{act}$ (*dpp-Gal4, UAS-GFP, UAS-Pvr*$^{act}$) induces LDH activity. (c–j) *LDH-GFP* transgene induction (shown in green) monitors the expression of LDH. Nuclei are marked with To-Pro (blue). Full genotype in each panel includes the driver (*dpp*$^{blk-1}$*-Gal4*), a *UAS*-transgene as indicated and mCherry to mark expressing cells (red channel omitted for clarity). (c) Control. No *LDH-GFP* expression is detected in *dpp-Gal4, UAS-mCherry* wing disc. (d) Pvr$^{act}$ (*dpp-Gal4, UAS-mCherry, UAS-Pvr*$^{act}$) causes robust *LDH-GFP* induction. (e–j) Either no expression or very mild expression (arrows in **e, h**) of *LDH* is seen when *Ras*$^{act}$ (**e**), *wingless* (**f**), *unpaired* (**g**), *yorkie* (**h**), *InR*$^{act}$ (**i**), or *Egfr*$^{act}$ (**j**) transgenes are expressed under the control of *dpp-Gal4* in the wing disc.
The following figure supplement is available for figure 1:

**Figure supplement 1.** Endogenous LDH activity and expression in *Drosophila* larval tissues.

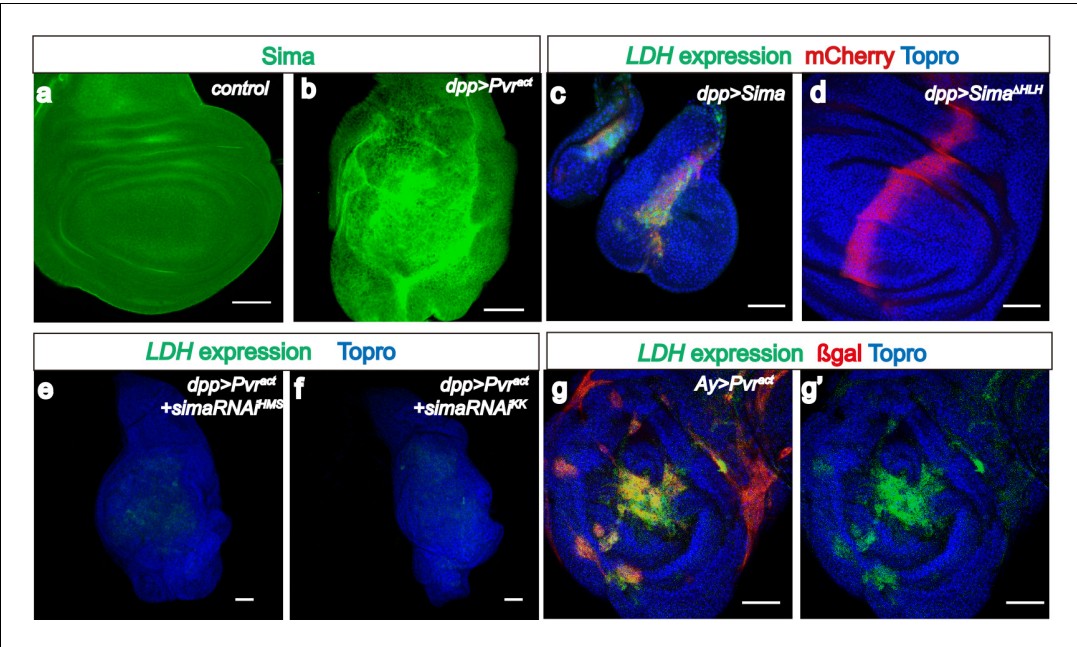

**Figure 2.** Sima mediates *LDH* expression. All wing discs shown are from wandering third instar larvae. All transgenes are driven using *dpp^{blk-1}-Gal4.* Scale bars, 50 µm. Nuclei are marked with To-Pro (blue). (**a–b**) Sima protein accumulation upon Pvr^{act} expression is detected with an α-Sima antibody (green). The expression domains are independently marked (not shown for clarity). (**c–d**) Direct overexpression of Sima protein (**c**), but not Sima lacking DNA-binding domain (**d**) leads to *LDH-GFP* (green) expression. mCherry (red) independently marks the zone of expression. (**e–f**) Two independent RNAi constructs against *Sima* (*simaRNAi^{HMS}* and *simaRNAi^{KK}*) can each suppress Pvr^{act}-induced expression of *LDH-GFP*. (**g–g'**) Ay-Gal4 induced small clones expressing Pvr^{act} (**g**, marked in red) autonomously induces *LDH-GFP* (green) (*hs-flp, UAS-lacZ, UAS-Pvr^{act}*).

The following figure supplement is available for figure 2:

**Figure supplement 1.** Sima (*Drosophila* Hif-α) regulates LDH activity and has moderate effects on tumor growth.

modest compared to that seen with activation of Pvr (*Figure 1e*). Together, the enzyme and the reporter assay establish that activation of PDGF/VEGF Receptor leads to transcriptional up-regulation of *LDH* and the subsequent formation of active LDH enzyme in the tumor tissue.

Expression of many oncogenes cause large tumorous overgrowths that superficially look like the Pvr induced tumors. However, *LDH* is not expressed in every tumor that shows increased cell proliferation and over-growth. For example, over-expression of the secreted ligand, Wingless (*Neumann and Cohen, 1996*), causes overgrowth and duplication of the tissue but does not cause *LDH* expression (*Figure 1f*). Similarly, overexpression of the JAK/STAT pathway ligand Unpaired (Upd), promotes tissue growth (*Chao et al., 2004*; *Rodrigues et al., 2012*), but does not induce *LDH* expression (*Figure 1g*). The transcription factor Yorkie is a potent growth-promoting signal in the wing disc (*Huang et al., 2005*), but at best, it induces very weak and variable expression of *LDH-GFP* in the wing pouch (*Figure 1h*). These results indicate that the up-regulation of LDH by Pvr is not a general consequence of stepped up cell proliferation seen in all tumors.

Furthermore, given Pvr's function as an RTK, it was initially a surprise to find that neither constitutively signaling form of the insulin receptor (InR^{act}, *Figure 1i*) nor the activated form of epidermal growth factor receptor (Egfr^{act}; *Figure 1j*) can induce *LDH-GFP* expression even though both cause significant overgrowth of the tissue. This observed specificity of Pvr, compared with other RTKs, in the expression of *LDH*, presents a unique opportunity for investigating the mechanism by which an oncogene can regulate the metabolism of a tumor tissue.

Hif-1α is a well-known inducer of LDHA expression in many cancer cell lines (*Dang and Semenza, 1999*). As with the mammalian counterpart, *Drosophila* Hifα (called Sima) is rapidly degraded under

normoxic conditions and therefore no Sima protein is detected in the wild-type wing disc (*Figure 2a*), but Sima protein is stabilized in a Pvr[act] background (*Figure 2b*). Overexpression of Sima (but not of a Sima mutant lacking its DNA binding domain) robustly induces both LDH activity and expression (*Figure 2c–d*; *Figure 2—figure supplement 1a*). It also causes growth defects, similar to the phenotypes of the *Hph* mutant (*Centanin et al., 2005*). These growth defects associated with gross overactivity of Sima are likely independent of its role in hypoxia related responses.

Importantly, *LDH* expression induced by Pvr[act] is strongly suppressed upon silencing *sima* using either one of two independent transgenic RNAi constructs (*Figure 2e–f*). Thus, Sima is essential for LDH induction by Pvr[act]. Knockdown of *sima* does suppress some of the overgrowth caused by Pvr[act], but this effect is small (about a 20% reduction in volume. see *Figure 2—figure supplement 1b*) and the tumor-growth loss is clearly not as robust as the complete loss of *LDH-GFP* seen using the same knockdown construct. We conclude that tumor growth is a result of multiple interacting events, and blocking glycolysis alone will not be sufficient to fully rescue the growth of a tumor.

The Pvr induced tumor is characterized by unrestrained, disorganized growth, and in principle, one mechanism for Sima stabilization could involve the hypoxic environment that might exist within the disorganized mass of a tumor tissue. Examples of such a mechanism in which deeper tissues within solid tumors become hypoxic have been demonstrated in human cancers (*Bertout et al., 2008*; *Keith and Simon, 2007*). However, it is also true that in addition to hypoxia, Hif can be stabilized under normoxic conditions by multiple metabolites such as NO, ROS, succinate, and fumarate (*Chandel et al., 2000*; *Isaacs et al., 2005*; *Selak et al., 2005*; *Mateo et al., 2003*). To address whether the Pvr induced *LDH-GFP* expression is specifically due to a hypoxic core within a tumor, we utilized the *AyGal4* system (*Ito et al., 1997*) to generate flip-out clones expressing Pvr[act]. This technique allows analysis of *LDH* expression in single or small groups of cells expressing Pvr[act] that are surrounded by normal tissue. Under these conditions, there is no over-growth or tumor formation, yet these single/small number cell clones express *LDH* in a strikingly cell-autonomous fashion (*Figure 2g*). We conclude that a large tumor with a hypoxic core is not a requirement for Sima stabilization or *LDH* expression.

## Both ERK and PI3K pathways are necessary for LDH expression

Ras is the major effector of RTK signaling (*Moodie et al., 1993*; *Vivanco and Sawyers, 2002*), and indeed we have found that constitutively active dRas1 is sufficient for a small increase in *LDH* expression and activity (*Figure 1e*). Also, induction of *LDH* by Pvr[act] is suppressed by a dominant-negative mutant allele of *Ras (dRas1^{N17})* (*Lee et al., 1996*) (*Figure 3h*). Interestingly, although Ras is downstream of both the EGFR and InR (Insulin receptor) pathways, on their own, neither Egfr[act] nor InR[act] causes *LDH* expression even as they individually cause overgrowth (*Figure 1i–j*). We found that Egfr[act] causes phosphorylation of ERK but not AKT (*Figure 3a,d*), while InR[act] activates Akt but not ERK (*Figure 3b,e*). In this system, Pvr[act] induces both pathways (*Figure 3c,f*) and thus we tested the function of each pathway and then both in combination in the control of *LDH* expression. RNAi initiated knock down of Dsor1 (MAPKK) or ERK (RI, *rolled*), potently suppresses *LDH-GFP* induction downstream of Pvr[act] (*Figure 3i–j*). Thus, ERK pathway is essential for Pvr-mediated *LDH* upregulation.

Similarly, when we blocked PI3K signaling in a Pvr[act] background using several independent ways, including co-expression of an RNAi against PI3K (*Figure 3k*), a dominant-negative mutant form of PI3K (*Figure 3l*), inactivation of Akt by RNAi (*Figure 3m*), or silencing dTOR by RNAi (*Figure 3n*), each combination suppresses the induction of *LDH-GFP*. Finally, Pvr[act] dependent accumulation of Sima protein is significantly suppressed when either Dsor1 or Akt is silenced (*Figure 3—figure supplement 1a–b*). Thus, the PI3K/Akt/TOR axis is also definitively needed for LDH regulation.

Importantly, individual overexpression of either an activated form of human Raf (hRaf[act]) or a constitutively active form of PI3K (PI3K[act], active due to a mutation in the p110α catalytic subunit) (*Brand and Perrimon, 1994*; *Leevers et al., 1996*) does not induce *LDH-GFP* expression although each genotype causes significant over-growth (*Figure 3o–p*). In combination, however, co-expression of hRaf[act] and PI3K[act] either using drivers or in small clones, leads to extensive *LDH* induction even more robustly than what is seen for Pvr[act] (*Figure 3q–r*). Furthermore, dual PI3K/ERK activation is also sufficient to increase Sima expression (*Figure 3—figure supplement 1c–d*). Knock down of Sima significantly attenuates PI3K/ERK induced *LDH-GFP* expression (*Figure 3—figure supplement 1h–i*). Similar to its role in mediating Pvr[act] induced *LDH* expression, this demonstrates that Sima is

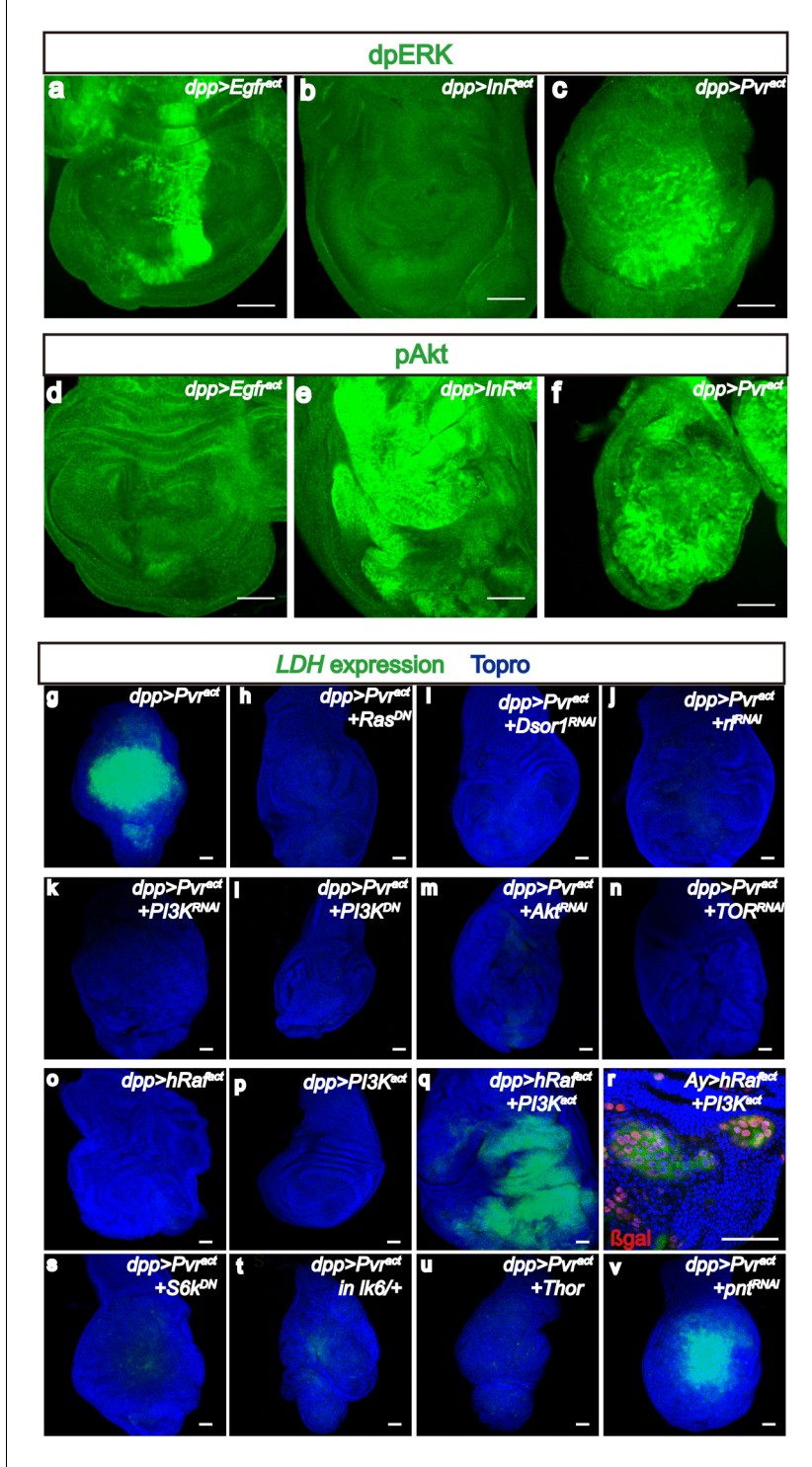

**Figure 3.** Downstream of Pvr[act], ERK and PI3K pathways control Sima translation. All wing discs shown are from wandering third instar larvae. All transgenes are expressed using the *dpp[blk-1]-Gal4* driver. Scale bars, 50 μm. (a–f) Differential activation of ERK and PI3K pathways by RTKs. Phospho-specific antibodies against dp-ERK (a–c, green) or p-Akt (d–f, green) are used to detect activation of the corresponding pathways. (a–c) dp-ERK staining is seen in in Egfr[act](a) and Pvr[act](c) backgrounds, but not upon activation of InR (b). (d–f) In contrast to (a–c), p-Akt is detected upon activation of InR (e) and Pvr (f) but not Egfr (d). Thus only Pvr[act] causes both the ERK (assayed by ERK phosphorylation) and PI3K (assayed by Akt phosphorylation) pathways to be activated. (g–v) Both PI3K and ERK pathways must be activated for *LDH-GFP* induction. Expression of *LDH-GFP* (green) marks glycolytic tissue. In

*Figure 3 continued on next page*

*Figure 3 continued*

(r), the *Ay-Gal4* system is used to generate clones with nuclear β-Galactosidase staining (red) marking the clones. *LDH-GFP* (green) expression (g; Pvr$^{act}$ control) is suppressed upon loss of ERK pathway members Ras (using a dominant negative, h), Downstream of Raf1, (Dsor1, using an RNAi construct, i), and ERK (Rolled, using an RNAi construct, j). (k–n) PI3K pathway components are also essential for *LDH-GFP* expression. *LDH-GFP* (green) expression (g; Pvr$^{act}$ control) is suppressed upon loss of PI3K pathway members Ras (using a dominant negative, h), PI3K (using an RNAi construct, k or a dominant negative version, l), Akt (using an RNAi construct, m), or TOR (RNAi; n). (o–r) ERK and PI3K pathways must be co-activated for *LDH* expression. (o–p) Single activation of the ERK pathway (using hRaf$^{act}$; o), or of the PI3K pathway (using PI3K$^{act}$; p) is insufficient for *LDH-GFP* expression. (q,r) Co-activation of both pathways by co-expression of hRaf$^{act}$ and PI3K$^{act}$ together, either using a *dpp-Gal4* driver (q) or in small *Ay-Gal4*-derived clones (r) induces robust *LDH-GFP* expression. (s–v) Translational regulators are essential for *LDH* transcriptional control. A dominant-negative form of S6k (s) or overexpression of Thor/4EBP (u), both acting downstream of PI3K as well as loss of a single copy of Mnk/Lk6 (t), functioning downstream of ERK, abolish transcription of *LDH-GFP*. In contrast, loss of the major transcriptional factor in the ERK pathway, the ETS domain protein, Pointed (v) has no effect on *LDH* transcription.

The following figure supplement is available for figure 3:

**Figure supplement 1.** PI3K and ERK pathway members are required for Sima accumulation.

the critical regulator of glycolytic genes upon co-activating PI3K and ERK pathways. We conclude that a combination of ERK and PI3K signaling is both necessary and sufficient to induce Hifα-dependent *LDH* induction. On its own, each signal is necessary but not sufficient to cause *LDH* expression.

InR/TOR pathway is associated with translational control in diverse systems (*Fukuda et al., 2002*; *Hay and Sonenberg, 2004*). In particular, both translation and degradation critically control Hifα level (*Bacon et al., 1998*; *Gorr et al., 2004*). Consistent with this notion, the induction of *LDH* by Pvr$^{act}$/Sima remains unaffected when Pointed (*Drosophila* ETS2), the well–established direct nuclear target of the ERK pathway (*O'Neill et al., 1994*), is mutated (*Figure 3v*). In contrast, the ribosomal S6 kinase, a known target of TOR signaling that promotes translation of a subset of mRNA transcripts, including Hif-1α (*Ma and Blenis, 2009*) is prominently involved in this process since a dominant-negative form of S6k is a strong suppressor of Pvr-mediated *LDH* induction and Sima accumulation (*Figure 3s* and *Figure 3—figure supplement 1f*). The ERK signal also modulates protein translation through the Mnk kinase (LK6 in *Drosophila*). Lk6 binds ERK and phosphorylates the initiation factor eIF4E to promote Cap dependent translation (*Arquier et al., 2005*; *Parra-Palau et al., 2005*). Strikingly, we find that a single-copy loss of *lk6* strongly suppresses *LDH* induction and Sima accumulation downstream of Pvr (*Figure 3t* and *Figure 3—figure supplement 1g*). Finally, published literature (*Miron et al., 2001*) has established 4E-BP (Thor) as a downstream effector of the PI3K/Akt pathway that forms a complex with eIF4E and inhibits translation. mTOR phosphorylates 4E-BP releasing it from eIF4E (*Furic et al., 2010*) allowing efficient translation. We found that co-expression of 4E-BP with Pvr$^{act}$ suppresses *LDH* induction (*Figure 3u*). Taken together, these data strongly support a mechanism in which ERK and PI3K pathways converge at the level of translational control of gene products including Sima. We propose that initially, the increased translation of *sima* transcript generates sufficient Sima protein (in excess of the rate of Sima degradation), leading to *LDH* induction; later we will show that this reprogramming is further augmented by a feedback loop involving ROS.

## Pvr$^{act}$ induced metabolic changes

### Intensification of glycolytic pathway

In order to gain better understanding of the transcriptional basis of Pvr-induced glycolytic transition at a genome-wide level and the role of *sima* in the regulation of this process, we carried out an RNA-Seq experiment in which transcriptomes of wild-type wing imaginal discs were compared with those with the genotypes *Pvr$^{act}$*(glycolytic activity), or *InR$^{act}$*(no detectable glycolytic activity) or *Pvr$^{act}$ + sima$^{RNAi}$* (glycolytic activity suppressed). Each genotype was analyzed in triplicate. The data were filtered for the GO term 'metabolism', and further sorted for genes that are up-regulated in Pvr$^{act}$ but not in InR$^{act}$ background.Amongst this subset, transcripts that are down-regulated in

Pvr[act] + sima[RNAi] background were analyzed further. Our results confirm the transcriptional up-regulation of LDH. Interestingly, the transcription of six of the ten glycolytic enzymes is up-regulated in a Pvr[act] background of which four –Hex-A, Pfk, Ald and Impl3- are regulated in a Sima dependent manner (*Figure 4a–b*). The 73 *sima*-dependent metabolic genes identified in this assay are listed in *Figure 4—figure supplement 1*. GO and network analysis reveal enrichment in biological function for genes involved in regulating rate-limiting steps of glycolysis. Hex-A (hexokinase-A) phosphorylates glucose to generate glucose 6-p in the first step of glycolysis. Similarly, Pfk (phosphofructo kinase) phosphorylates fructose 1-p to generate fructose 1,6-bp, whereas Ald (Aldolase) converts fructose 1,6-bp to glyceraldehyde 3-p in the next step. Impl3/LDH (lactate dehydrogenase) converts pyruvate to lactate in the final step of glycolysis. Pvr and Sima-dependent regulation of these key enzymes indicates a more efficient usage of the entire glycolytic pathway.

## Attenuation of oxidative phosphorylation

In addition to displaying increased aerobic glycolysis, a subset of cancer cells also attenuate mitochondrial respiration (*Pelicano et al., 2006*; *Warburg, 1956a*, *1956b*). The RNA-seq data identifies a set of transcripts encoding mitochondrial proteins that is down-regulated in Pvr[act] background. Attenuation of ETC complex protein activity has also been demonstrated to up-regulate transcripts for all glycolytic enzymes suggesting a cross talk between the two primary modes of metabolism (*Owusu-Ansah et al., 2008*).

Just as LDH drives glycolysis by converting pyruvate to lactate, pyruvate dehydrogenase (PDH) drives the TCA cycle and oxidative phosphorylation by converting pyruvate to acetyl-CoA. PDH is inactive when phosphorylated by p-PDHK. When stained with appropriate antibodies, Pvr[act] cells are found to express PDHK[total], active PDHK (phospho-PDHK), as well as the inactive form of PDH (p-PDH) (*Figure 5a–f*). The high p-PDH level is suppressed by co-expressing PDHK-dsRNA (*Figure 5—*

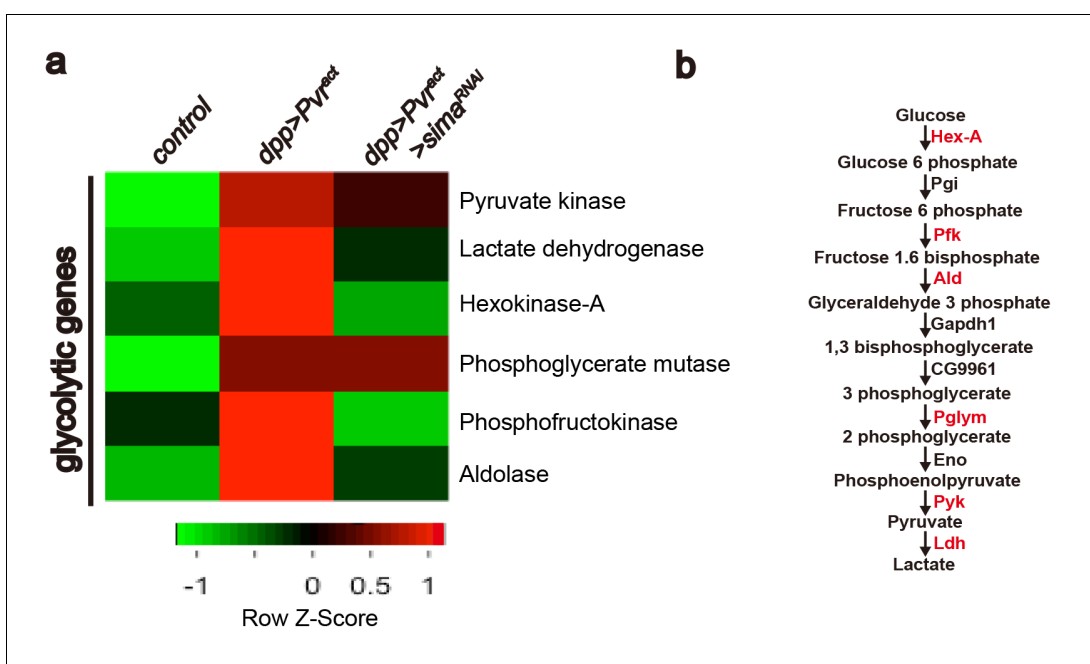

**Figure 4.** Heat map of *sima*-dependent glycolytic gene expression. An unsupervised hierarchical cluster heat map based on differential mRNA expression patterns yielded 73 *sima*-dependent genes responding to the GO term analysis as metabolic genes (see *Figure 4—figure supplement 1*). This includes transcripts for six up-regulated enzymes (marked in red) that belong to the glycolytic pathway (**b**). RNAs for four glycolytic enzymes are induced by Pvr[act] in a *Sima* dependent manner (**a**).

The following figure supplement is available for figure 4:

**Figure supplement 1.** Expression of 73 metabolic genes are up-regulated by Pvr[act] in a Sima dependent manner.

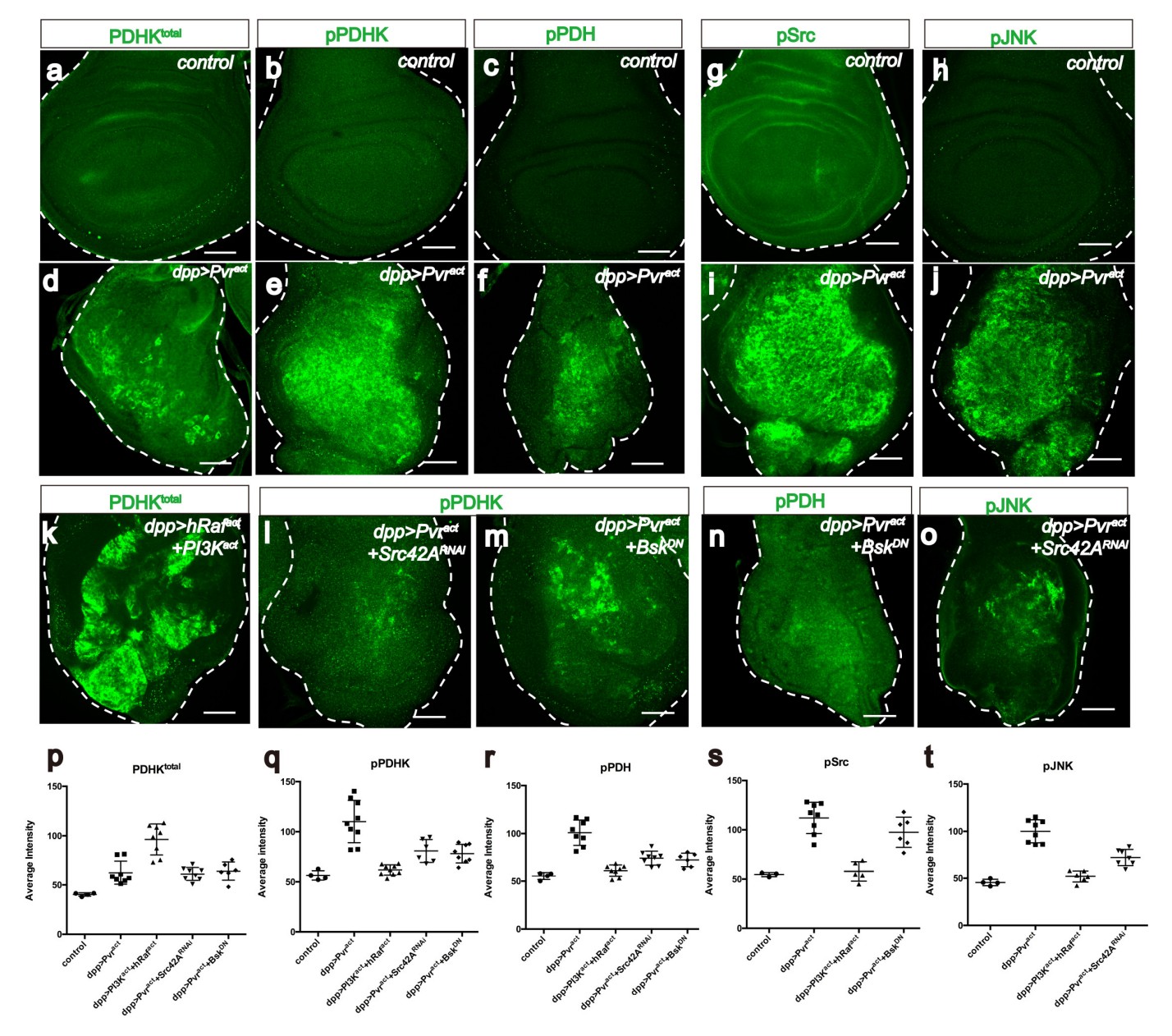

**Figure 5.** Src42A and JNK inhibit mitochondrial function by regulating PDHK activity. All wing discs shown are from wandering third instar larvae. All transgenes are expressed using the *dpp^{blk-1}-Gal4* driver. Scale bars, 50 μm. (**a–f**) Pvr^{act} up-regulates PDHK expression as well as its activity. Total PDHK protein expression detected using an α-PDHK1 antibody (**a, d**, green) while phospho PDHK-specific antibody detects only the activated form (**b, e**, green). p-PDH (**c, f**, green) is detected only if PDH is phosphorylated and inactivated by p-PDHK. Compared to control, Pvr^{act} tissue shows elevated expression of PDHK^{total} (**a, d**), p-PDHK (**b, e**), and p-PDH (**c, f**). (**g–j**) Pvr^{act} activates Src and JNK. Compared with controls, Pvr^{act} tissue shows increased expression of activated Src ((**g, i**) phospho Src specific, green) and activated p-JNK ((**h, j**) phospho JNK specific, green). (**k**) Co-activation of ERK and PI3K pathways dramatically increases PDHK^{total} (green) expression, but this PDHK is inactive (see Supplementary *Figure 5c*). (**l–o**) Src and JNK activate PDHK, which in turn inactivates PDH. p-PDHK (green) is suppressed upon loss of Src42A (using RNAi; **l**), or JNK (using a dominant-negative version Bsk^{DN}; **m**). (**n**) Reduction in JNK signaling reduces p-PDH (**o**) Reduction in Src42A suppresses p-JNK (green) but loss of JNK does not affect p-Src (**s**) suggesting that JNK functions downstream of Src42A. (**p–t**) Quantitative analysis of p-PDHK, p-PDH, p-Src, p-JNK expression in various genotypes, indicated in each graph. Error bars indicate standard error of the mean with significance determined by t-test. (**p**) Activation of either Pvr or a combination of ERK and PI3K pathways causes translational up-regulation of PDHK protein (p=0.0045 and 0.0001), independent of Src and JNK (p=0.7864 and 0.7935). (**q**) Activation of PDHK by phosphorylation required Src and JNK function (p=0.0086, 0.0005 and 0.0013) downstream of Pvr, but is independent of the PI3K and ERK pathways (p=0.0978). (**r**) Src and JNK function downstream of Pvr (p=0.0002 and 0.0005), independent of PI3K/ERK pathway (p=0.1201) to inactivate PDH. (**s**) Pvr activates Src (p=0.0002). This regulation is independent of ERK/PI3K pathways (p=0.6064). (**t**) JNK

*Figure 5 continued on next page*

*Figure 5 continued*

phosphorylation requires Src (p=0.0002) but Src phosphorylation does not require JNK (p=0.1117). The regulation of JNK is independent of the ERK and PI3K pathways (p=0.0855).

The following source data and figure supplement are available for figure 5:

**Source data 1.** The numerical data for *Figure 5p–t*.

**Figure supplement 1.** Pvr[act] induces PDHK translation through ERK and PI3K pathways.

*figure supplement 1a*) establishing that PDHK is indeed required for inactivation of PDH in Pvr[act] cells.

An antibody that recognizes both the active and the inactive forms of PDHK shows that co-expression of hRaf[act] and PI3K[act] results in dramatic up-regulation of PDHK[total]. Similar to the results obtained for Sima, the regulation of PDHK protein is also at a translational level (*Figure 5k*; *Figure 5—figure supplement 1o,q*). Interestingly, while the combined activation of PI3K[act] and hRaf[act] is sufficient for PDHK production, the protein thus produced is inactive and is not recognized by a phospho-specific, p-PDHK antibody (*Figure 5—figure supplement 1c*). This is unlike in Pvr[act] cells where both PDHK[total] and PDHK[active] are seen (*Figure 5d–e*). Consistent with this observation, p-PDH (the inactive form of PDH) is also not seen when hRaf[act] and PI3K[act] are co-expressed (*Figure 5—figure supplement 1b*), and yet is readily detectable in a Pvr[act] background (*Figure 5f*). These results establish that additional proteins independent of ERK and PI3K pathways function downstream of Pvr and are necessary to activate PDHK and thus inhibit PDH and mitochondrial oxidative phosphorylation.

Src42A, a *Drosophila c-src* proto-oncogene homolog, functions in epidermal closure during both embryogenesis and metamorphosis by regulating JNK signaling (*Tateno et al., 2000*). It has also been found that Src42A controls tumor invasion and cell death by activating JNK (*Ma et al., 2013*). In mammalian cancer studies, Src mediates tumor cell metastasis and angiogenesis induced by VEGF (*Eliceiri et al., 1999*; *Weis et al., 2004*). Using phospho-specific antibodies, we found that both Src42A and JNK are extensively phosphorylated in Pvr[act] cells (*Figure 5g–j*). Knockdown of Src42A suppresses phospho-JNK accumulation induced by Pvr[act] (*Figure 5o*), whereas over-expression of a dominant negative form of JNK (Bsk[DN]) does not significantly reduce phospho-Src levels (*Figure 5s*) indicating that Src42A functions upstream of JNK.

Knockdown of Src42A, or over-expression of Bsk[DN] partially suppresses the phosphorylation level of PDHK induced by Pvr[act] (*Figure 5l–m*). As shown earlier, co-expression of hRaf[act] and PI3K[act] is unable to increase p-PDHK levels (*Figure 5—figure supplement 1c*), but the simultaneous activation of ERK, PI3K and JNK pathways by using hRaf[act], PI3K[act] and active JNKK (hemipterous, hep[act]), clearly up-regulates PDHK activity (*Figure 5—figure supplement 1d*), and the resulting effect on PDH is also dramatic. A dominant negative form of JNK (Bsk[DN]) or loss of JNKK (hemipterous, hep[R-NAi]) suppresses the formation of p-PDH upon Pvr activation (*Figure 5n*; *Figure 5—figure supplement 1e*). Thus JNK is required for activation of p-PDHK and as a result, sufficient to inactivate PDH by causing its phosphorylation.

In summary, the above results show that:

1. Src functions downstream of Pvr, independent of the PI3K/ERK pathway (*Figure 5p*; *Figure 5—figure supplement 1g*)
2. JNK functions downstream of Pvr and Src, independent of the PI3K/ERK pathway (*Figure 5q, s*; *Figure 5—figure supplement 1h*)
3. Activation of the ERK/PI3K pathways downstream of Pvr, causes translational up-regulation of the PDHK protein; neither Src nor JNK has a role in controlling total PDHK protein level (*Figure 5p*; *Figure 5—figure supplement 1i,j,q*)
4. The PDHK protein produced by the PI3K/ERK pathway is catalytically inactive. The activation process produces phospho-PDHK and involves the Src/JNK pathway (*Figure 5l,m,q*)
5. p-PDHK is necessary for inactivating PDH by phosphorylation. It follows that the process of phosphorylation dependent PDH inactivation is downstream of Src and JNk (*Figure 5r*; *Figure 5—figure supplement 1e,f*).

## Feedback loops to strengthen the metabolic reprogramming

Although over-active PDH can promote reactive oxygen species (ROS) production during high mitochondrial flux (*Kaplon et al., 2013*), dysfunction of PDH within the mitochondrion is invariably associated with increased production of ROS (*Glushakova et al., 2011*; *Ambrus et al., 2011*). Thus, Pvr[act] cells exhibit high ROS levels detected by direct staining with the DHE dye (*Figure 6a–b*) or as monitored by *gstD-GFP* expression (*Robinson et al., 2006*) (*Figure 6c–d*) frequently used as a surrogate for ROS, marking cells under oxidative stress (*Ohsawa et al., 2012*; *Sykiotis and Bohmann, 2008*). This ROS induction is suppressed when PDHK[RNAi] is co-expressed (*Figure 6—figure supplement 1a*).

Co-expression of hRaf[act] and PI3K[act] does not cause ROS production unlike that seen for Pvr[act] (*Figure 6e*; *Figure 6—figure supplement 1b*). Instead, Pvr[act] induced ROS is effectively suppressed in a Bsk[DN] (JNK dominant negative) genetic background (*Figure 6f*; *Figure 6—figure supplement 1c*). Similar effects are seen upon knockdown of hep (JNKK) or Src42A (*Figure 6—figure supplement 1d–e*). Thus Src-JNK signaling plays an important role in the inhibition of mitochondrial activity and in raising ROS levels in Pvr[act] tumors.

Two notable facts about ROS have a bearing on the data presented here. The first is that ROS can activate the JNK pathway through interaction with the upstream component ASK (JNKKK) (*Tobiume et al., 2001*; *Owusu-Ansah and Banerjee, 2009*). The second is that high ROS conditions result in potent stabilization of the Hif protein even under normoxic conditions (*Chandel et al., 2000*; *Fandrey et al., 2006*). This provides the opportunity for ROS generated downstream of both Hif and JNK, to feedback and reinforce these two pathways. To test this hypothesis, several antioxidant (free radical scavenging) proteins were expressed to reduce the level of ROS. Of these, the strongest ROS scavenging activity was seen upon Peroxidasin (Pxn) expression. In addition to a decrease in ROS (*Figure 6g–h*), we found that JNK is no longer activated when Pxn is co-expressed in Pvr[act] discs (*Figure 6i–j*). Similarly, *LDH-GFP* expression (*Figure 6k–l*) and the accumulation of Sima protein are strongly suppressed by co-expression of Pxn (*Figure 6m–n*). These results establish ROS as the central player in enforcing the metabolic reprogramming. Initially established through the activation of three oncogenic pathways by Pvr, the shift to glycolysis is then reinforced when ROS is generated as a subsequent step. This establishes one means to maintain a stable Warburg effect by oncogene activation.

## Discussion

A model (*Figure 7*) can be constructed that is consistent with all our observations on the Pvr[act] induced interacting network that leads to aerobic glycolysis and potentially also allows this new metabolic state to be maintained through later stages of tumor growth. This is a genetic model derived from analysis that allows placement of gene function according to their hierarchy along a pathway. Many genetic backgrounds that achieve full or partial Warburg effect have been described in the literature (*Elstrom et al., 2004*; *Kim and Dang, 2005*; *Wang et al., 2011*; *Faubert et al., 2013*; *Hitosugi et al., 2009*, *Zhang et al., 2013*, *Yang et al., 2012*; *Hong et al., 2016*; *Jin et al., 2016*). Although the details may seem to vary, the three critical components involved in this process are the two pyruvate-metabolism enzymes, LDH and PDH and the free radical metabolite class designated as ROS. During the acquisition of the aerobic glycolytic activity in a tumor environment, LDH actively converts pyruvate to lactate, and high PDHK inactivates PDH resulting in low pyruvate to acetyl-CoA conversion, low TCA flux and electron transport activity. Our results suggest that these events can be sustained over long periods of tumor growth only when coupled with a feed back signal from accumulating ROS in a mechanism that reinforces and enhances the high-LDH/low-PDH activity state. This genetic study was achieved in an in vivo context of an animal that is only mutant for the one specific activated oncogene that we introduce and is otherwise normal. Additional pathways are activated without accumulating new mutations. Also, while hypoxic conditions might favor a glycolytic activation in general, hypoxia is not an essential component for a tumor to become glycolytic, as in this system, where Hif is stabilized by alternative means in a normoxic environment.

Pvr[act] triggers two parallel pathways in a Ras dependent manner. The first is the PI3K/Akt pathway that targets the S6K protein important for translational initiation. The second is the Raf/ERK

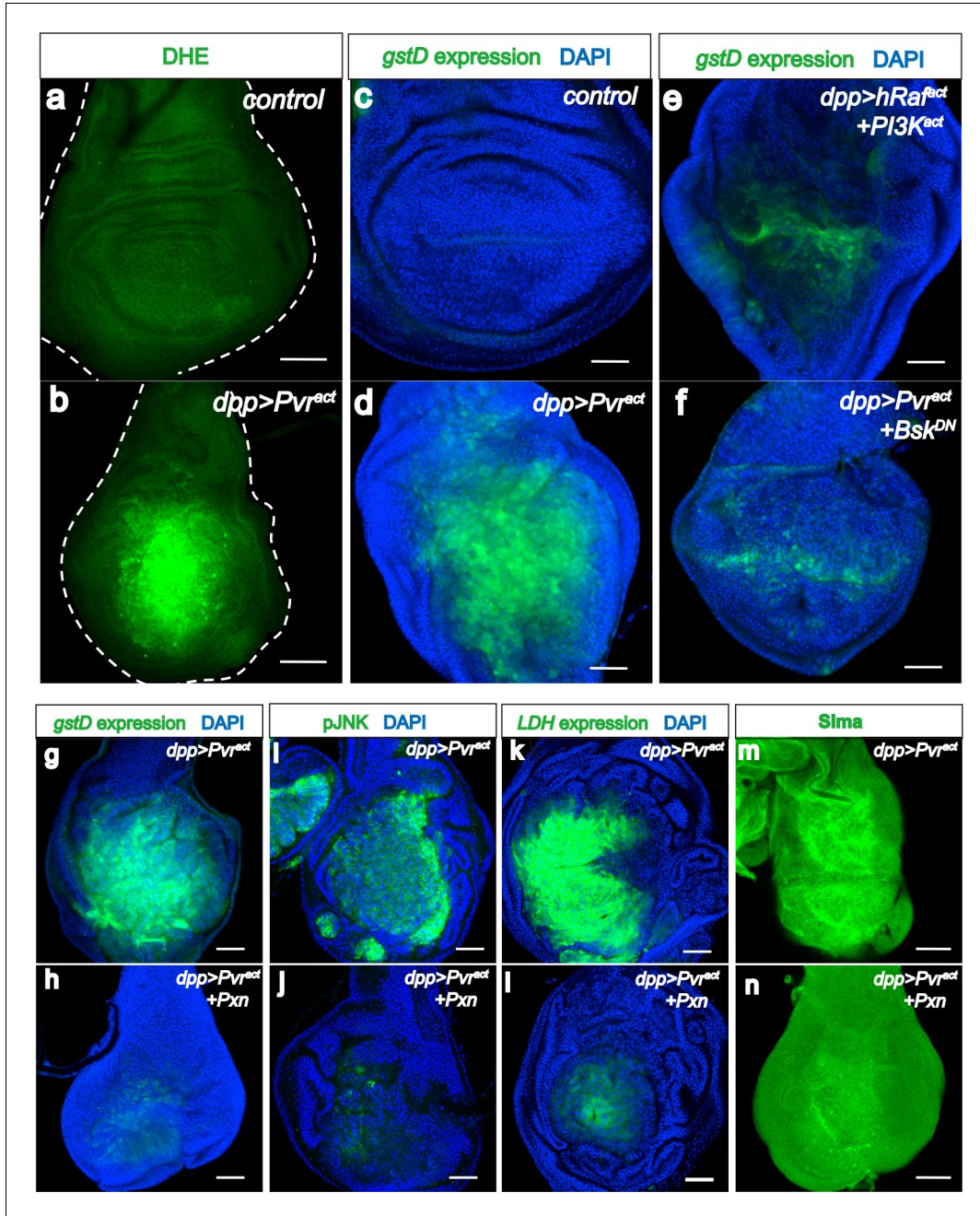

**Figure 6.** ROS strengthens the metabolic reprogramming. All wing discs shown are from wandering third instar larvae. All transgenes are expressed using the *dpp$^{blk-1}$-Gal4* driver. Nuclei are marked with DAPI (blue; **c–l**). Scale bars, 50 μm. (**a–f**) JNK mediates ROS induction. DHE staining (green; **a**, **b**) directly monitors ROS levels and *gstD-GFP* expression (green; **c–f**) is a measure of oxidative stress caused by the generated ROS. No detectable ROS stress is seen in control discs (**a**, **c**), while Pvr$^{act}$ causes robust ROS and *gstD* generation (**b**, **d**). Co-expression of hRaf$^{act}$ and PI3K$^{act}$ does not phenocopy Pvr$^{act}$ in causing oxidative stress (**e**). A dominant negative form of JNK (Bsk$^{DN}$) effectively suppresses ROS generation (**f**). (**g–m**) A ROS feedback signal plays a central role in Pvr$^{act}$ induced metabolic reprogramming. Sima accumulation is detected by using an α-Sima antibody (**m–n**). Pvr$^{act}$ induced ROS generation (**g**), p-JNK expression (**i**), *LDH* expression (**k**), and Sima accumulation (**m**), are all strongly suppressed by expressing a scavenger gene, Peroxidasin (Pxn **h**, **j**, **l** and **n**).

The following figure supplement is available for figure 6:

**Figure supplement 1.** PDHK and JNK are required for generating ROS in Pvr$^{act}$ cells.

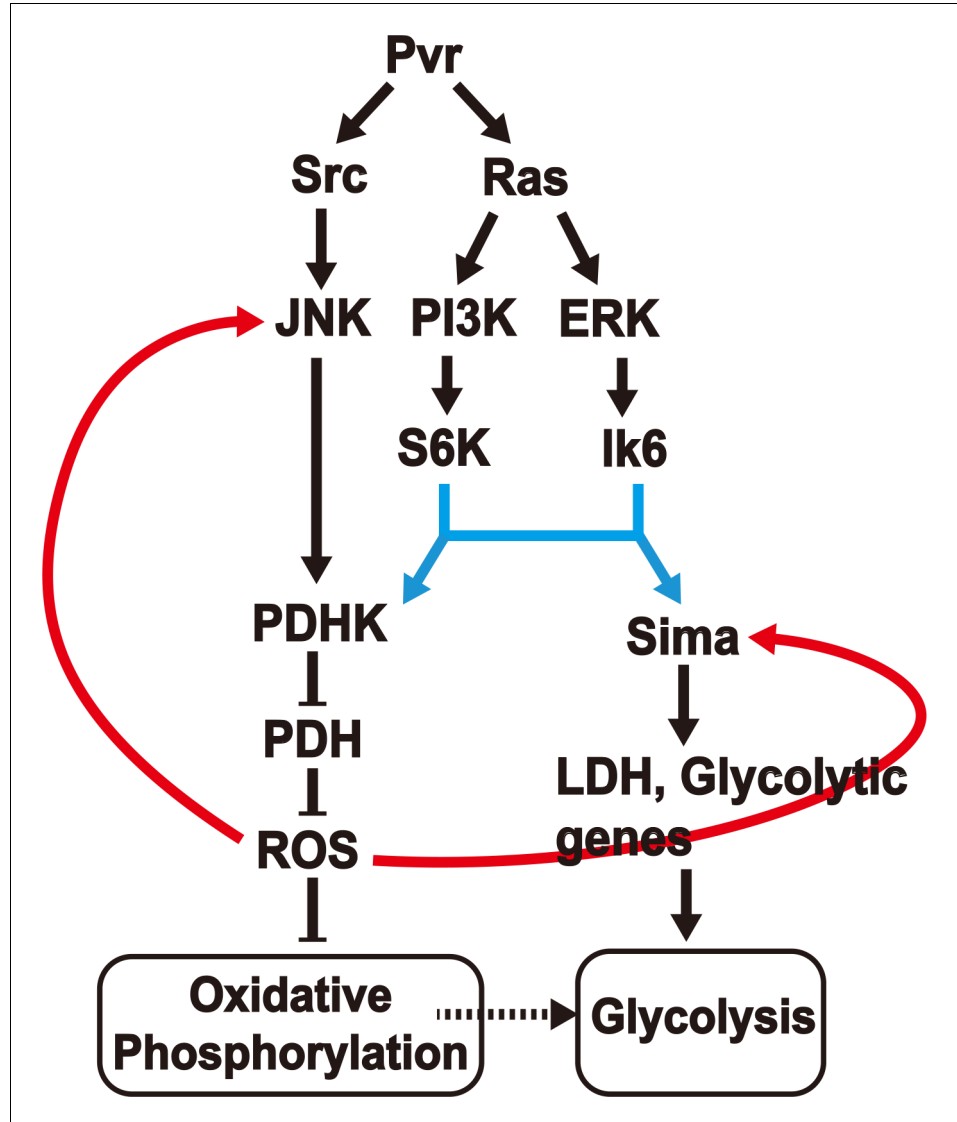

**Figure 7.** A model for induction and maintenance of a metabolic reprograming. Pvr[act] induced tumor tissues show a robust metabolic reprogramming from oxidative phosphorylation to aerobic glycolysis. The model describes the relationship between many pathways that are activated in response to the Pvr[act] signal. ROS, generated as a consequence of the initial metabolic reprogramming feeds back to upstream components, to consolidate and maintain the Warburg effect. See text for details.

pathway targeting the L6K (MNK in mammals) protein that is also independently required for translational initiation. On their own, any one of these pathways can cause tumor growth, presumably due to independent downstream transcriptional events. But both pathways must act simultaneously in order to establish translational control of both Hifα (Sima), and PDHK. Interestingly, while the PI3K and ERK pathways acting together causes significant amounts of PDHK translation, the protein thus produced is inactive. In contrast, the PDHK translated in a Pvr[act] background is enzymatically active and capable of inactivating PDH. Our data show that yet another parallel pathway initiated by Pvr[act] that involves Src and JNK helps generate the active form of PDHK. Therefore all three downstream effectors of activated Pvr (Src/JNK, PI3K/S6K and ERK/Lk6, eIF4E) are essential in generating an active form of PDHK protein to down-regulate oxidative phosphorylation, while only the last two of these pathways are needed for LDH production and glycolysis. It is easy to visualize how, in the context of the triggering oncogene, the tumor caused could either be non-glycolytic, or be glycolytic

but not lacking in oxidative phosphorylation (*Pavlova and Thompson, 2016*). In the system described in this paper, it is Inevitable that direct pyruvate to acetyl-CoA conversion within the mitochondrion, and therefore the TCA flux will be attenuated since PDH function is inhibited by PDHK in the tumor cells. However, we do not have direct evidence for or against any bypass mechanisms such as those using glutamine as the primary driver of anabolic processes being operational in the context of the *Drosophila* Pvr^act induced tumors as they are for many cancers. This issue requires further direct measurement of metabolic activity of the tumor tissue in the near future.

While this sequence of events can initiate a switch in the metabolic profile of the cell, an additional mechanism is required to sustain this transition over a long period of time. The central player in this sustenance is ROS. The level of these free-radical species rises as the mitochondrion becomes dysfunctional (*Sun et al., 2009*; *Glushakova et al., 2011*; *Owusu-Ansah et al., 2008*). Importantly, this excess ROS functions as a feedback signal that has at least two important consequences. First, ROS stabilizes Hif using the same mechanism that is used during hypoxia (*Chandel et al., 2000*; *Fandrey et al., 2006*), thus reinforcing the glycolytic pathways. The second is that ROS can activate the JNK pathway (*Tobiume et al., 2001*; *Owusu-Ansah and Banerjee, 2009*). Thus, after the initial reprogramming in metabolism, the transition is made stable over time, as the generated ROS reinforces the upstream members. All of the results shown here are consistent with this positive feedback model that enforces the Warburg effect and sustains it over time. For example, scavenging ROS blocks phosphorylation of components that are upstream of it (*Figure 6i–j*). Unlike Pvr, activation of PI3K and Raf together does not raise ROS even as it increases LDH (*Figure 3q* and *Figure 6e*); and activation of Ras, which will not lead to high ROS causes only very weak metabolic reprogramming compared with Pvr^act (*Figure 1d–e*). Thus, the positive feedback function of ROS is a critical step in tumors undergoing metabolic reprogramming or cell lines that are not necessarily under hypoxic conditions. Although tissue hypoxia does stabilize Hif (*Wenger, 2002*), this would be true of large non-vascularized solid tumors but not necessarily in circulating cells or in cancer cell lines. The mechanism detailed in *Figure 7* is Hif dependent but hypoxia independent, a condition often observed in human cancers (*Lee et al., 2008*; *Ivan et al., 2001*; *Zhong et al., 2000*; *Park et al., 2003*).

Consistent with translational control, transcript levels of PDHK or Sima are not altered in Pvr^act cells, whereas the PDHK and Sima protein levels are very significantly up-regulated by Pvr^act through the translational module controlled by PI3K/ERK pathways. Unlike in several published mammalian studies (*Kaplon et al., 2013*; *Kim et al., 2006*), we did not find evidence for transcriptional control of PDHK by Hif. Analysis of our RNA-Seq data reveals that PDHK transcription levels remain unaltered in Pvr^act background (*Figure 4—figure supplement 1*). This was also confirmed by quantitative RT-PCR (*Figure 5—figure supplement 1p*), The reason for this discrepancy is unclear; perhaps Hif does control basal transcriptional levels of PDHK, but the oncogene induced up-regulation of activity results from a post-transcriptional control of PDHK activity as we observe here.

It is interesting that the activation of a single oncogene, the PDGF/VEGF Receptor (Pvr) induces multiple phenotypes associated with tumor formation, including overgrowth, cell shape change, local migration and importantly for this study, a likely shift in metabolism from oxidative phosphorylation to aerobic glycolysis. These tumors are not metastatic presumably because this requires complete loss of epithelial polarity (*Pagliarini and Xu, 2003*; *Wu et al., 2010*). Although only a single oncogene is activated, the observed Warburg effect is caused by the activation of several interconnected pathways that are precisely dissected in this genetically tractable model system. The distinct advantage of this analysis is that the tumor is generated in an otherwise wild-type background such that the pathways activated reflect primary drivers of the different aspects of the tumor and the analysis is not complicated by background mutations. In some cancer cell-lines, a Warburg effect is often entirely attributed to PI3K activation (*DeBerardinis et al., 2008*). This interpretation is complicated by the presence of background mutations that make cell-lines immortal, before they are transformed. For example, PI3K activated tumors are often seen only in backgrounds in which either Raf, ERK or Ras is activated (*McCubrey et al., 2007*; *Steelman et al., 2008*; *Yuan and Cantley, 2008*). With the advent of modern gene manipulation technologies, it is now possible to create in vivo models in mouse (*Xue et al., 2014*; *Platt et al., 2014*), but determining the epistatic relationships between all genetic components continues to be a challenge. Given the complete conservation of all the relevant components between *Drosophila* and mammals, it is reasonable to propose that a similar set of epistatic relationships to the one proposed here, will hold in mammalian cancers.

## Materials and methods

### *Drosophila* stocks

The following fly stocks were used: $w^{1118}$, dpp-Gal4$^{blk-1}$, UAS- dRas1$^{V12}$(Ras$^{act}$), UAS-wg, UAS-upd, UAS-CD8:mCherry, Ay-Gal4 UAS-CD2, Ay-Gal4 UAS-lacZ, MKRS hsFLP.86E, UAS-dRas1$^{N17}$ (Ras$^{DN}$), UAS-S6k$^{KQ}$ (S6K$^{DN}$), lk6$^2$, UAS-cRaf1$^{gof}$, UAS-dp110$^{CAAX}$(PI3K$^{act}$), UAS-PI3K92E$^{A2860C}$ (PI3K$^{DN}$), UAS-InR$^{A1325D}$ (InR$^{act}$), UAS-Bsk$^{DN}$, UAS-hep$^{act}$, UAS-sima$^{RNAiHMS00832}$, UAS-Dsor1$^{RNAiHMS00710}$, UAS-Dsor1$^{RNAiHMS00145}$, UAS-Dsor1$^{RNAiJF03100}$, UAS-rl$^{RNAiHMS00173}$, UAS-PI3K92E$^{RNAiJF02770}$, UAS-Akt1$^{R-NAiHMS00007}$, UAS-dTOR$^{RNAiHMS00504}$, UAS-dTOR$^{RNAiHMS01114}$, UAS-PDHK$^{RNAiGL0009}$, UAS-Src42A$^{R-NAiHMS02755}$ (Bloomington *Drosophila* Stock Center, Bloomington, IN), UAS-Pvr$^{act}$ (λ-Pvr) (N. Perrimon), *UAS-yki (D. Pan), UAS-λEgfr* (T. Schupbach), *LDH-GFP (YD0852)* (L. Cooley), *UAS-sima$^{RNAiKK106187}$*(VDRC, Vienna, Austria), *UAS-sima* (P. Wappner), *gstD-GFP* (D. Bohmann)

### Immunohistochemistry

Standard protocols were used to fix and stain wing discs for immunohistochemistry. TO-PRO-3 or DAPI was used for a DNA counterstain (Invitrogen, Carlsbad, CA), and tissues were mounted in Vectashield (Vector Labs, Burlingame, CA). The following antibodies were used: anti-dpERK (1:100; Sigma-Aldrich, St. Louis, MO), anti-phospho *Drosophila* Akt (Ser505) (1:25; Cell Signaling, Danvers, MA), anti-Sima (1:100; generated by PRF&L, Canadensis, PA), anti-phospho PDH (S293) (1:100; Abcam, Cambridge, MA), anti-PDHK1 (1:100; Abcam), anti-phospho PDHK1 (Tyr243) (1:200; Cell Signaling), anti-phospho JNK (Thr183/Tyr185) (1:500; Cell Signaling), and anti-phospho Src (Tyr418) (1:200; Invitrogen). Single- plane fluorescence images were obtained using a Zeiss LSM 700 confocal microscope with Zen 2009 acquisition software. Images and average intensity of images were processed and measured using ImageJ. At least 20 discs were used for each genotype per experiment. Three technical replications were done for each experiment.

### Flip-out clones

The *Ay-Gal4* system (*Ito et al., 1997*), combined with a heat shock-driven FLP recombinase, was used to generate clones in which oncogenes and/or RNAi and other transgenes were expressed. Expressing cells were marked by including *UAS*-driven transgenes for rat CD2 or nuclear β- Galactosidase. A 30–45 min heat shock (37°C) was applied to larvae grown for approximately three days AEL (25°C), and larvae were dissected after approximately 48 hr of subsequent growth at 25°C.

### RNA-seq library construction and analysis method

Total RNA was isolated using PureLink RNA Mini Kit (Life Technologies, Carlsbad, CA) according to manufacturer's protocol. Total RNA fraction was processed using Illumina's ''TruSeq RNA Sample Preparation v2 Protocol'' (Illumina, San Diego, CA). The resulting purified cDNA library was sequenced on the Illumina Hiseq 2000 by following manufacturer's protocols. 50 bp single-end RNA-seq reads were obtained, and sequence files were generated in FASTQ format. The quality score of RNA-seq reads was obtained by using the FastQC. Reads were then aligned to the *Drosophila melanogaster* Reference Sequences UCSC dm6 from Illumina iGenome files using TopHat v2.0.10 (*Kim et al., 2013*).Transcript assembly and estimation of their abundances were calculated with Cufflinks 2.1.1 (*Trapnell et al., 2012*) by using the *Drosophila melanogaster* Reference annotation dataset UCSC dm6 from Illumina iGenome files (https://support.illumina.com/sequencing/sequencing_software/igenome.html). Differential expression for genes across the different conditions was calculated with Cuffdiff 2.1.1 (*Trapnell et al., 2012*). Heat maps showing gene expression levels (in FPKM, fragments per kilobase of transcript per million mapped fragments) through the different samples were drawn with package heatmap.2 in R studio for a subset of selected genes. Data deposited in ArrayExpress: E-MTAB-3808.

### Quantitative real-time PCR analysis

Total RNA was extracted from approximately forty 3rd instar larval wing discs using PureLink RNA mini kit (Ambion, Waltham, MA). The SuperScript III First-Stand synthesis SuperMix kit (Invitrogen) was used for first-stand cDNA synthesis. Relative quantitative PCR was performed by comparative $C_T$ method using Power SYBR Green PCR master mix kit (Applied Biosystems, Waltham, MA) and a

StepOne Real-Time PCR detection thermal cycler (Applied Biosystems). The levels of *RpL10* were used to normalize total cDNA input.

## Wing disc volume measurements and analysis

To synchronize larvae, eggs were collected on grape juice plates, and only larvae hatched within a three-hour period were used. After allowing newly hatched larvae to grow for 96 hr at 29°C, wing discs were dissected, fixed, washed, and stained with TO-PRO-3. Discs were then mounted in a drop of Vectashield and slowly moved to edge of drop (in increasingly less mounting medium) until they were stuck in position to the glass slide. Small, square coverslips were placed on either side of drop, and a larger, rectangular coverslip was placed spanning the smaller ones. All slides were fixed with nail polish. This mounting creates a gap (of coverslip thickness) between the top coverslip and slide, preventing the coverslip from pressing and deforming wing discs. When imaging, z-stacks were obtained with a distance between slices of 14 μm, and the entire disc was covered. To measure volumes, plugin 'A 3D editing' was used (ImageJ). Briefly, each disc was manually outlined in each z-section (using the channel for mCherry), and the plugin then calculated total volume. Brightness and contrast were enhanced identically for each sample to aid in outlining. At least 10 discs were used for each genotype per experiment.

## ROS assay

For ROS detection in wing discs, after incubation with DHE (Invitrogen) in RT, discs were rinsed twice in Schneiders medium, fixed for 5 mins in 4% Formaldehyde, and rinsed once with 1XPBS before confocal imaging.

## Acknowledgements

We would like to thank Wei Liao for his help with bioinformatic analysis of RNA expression data. We thank R Nagaraj for helpful comments and discussions, and the members of the Banerjee lab for support. We thank N Perrimon, D Pan, T Schupbach, L Cooley, P Wappner, D Bohmann, and the stock centers of Bloomington, the National Institute of Genetics (Tokyo, Japan), and the Vienna *Drosophila* RNAi Center (Vienna, Austria) for providing *Drosophila* stocks and reagents. KTJ was supported by a postdoctoral fellowship (#PF-10-130-01-DDC) from the American Cancer Society. This work is supported by National Institutes of Health (NIH) grant RO1-EY008152 to UB.

## Additional information

### Competing interests

UB: Reviewing editor, *eLife*. The other authors declare that no competing interests exist.

### Funding

| Funder | Grant reference number | Author |
|---|---|---|
| American Cancer Society | Postdoctoral fellowship (#PF-10-130-01-DDC) | Kevin T Jones |
| National Institutes of Health | RO1-EY008152 | Utpal Banerjee |

The funders had no role in study design, data collection and interpretation, or the decision to submit the work for publication.

### Author contributions

C-WW, Wrote the final paper, Acquisition of data, Analysis and interpretation of data, Drafting and revising the article; AP, Acquisition of data, Analysis and interpretation of data, Revising the article; KTJ, Acquisition of data, Initiating the draft; SKT, Acquisition of data; UB, Supervised the project, Wrote the final paper, Conception and design

### Author ORCIDs

Utpal Banerjee, http://orcid.org/0000-0001-6247-0284

## Additional files

**Supplementary files**

• Supplementary file 1. RNA-seq result indicating comparative gene expression values between different genotype. Gene ID, values of gene expression level of different genotype (Dcl for the *dpp>mCherry* control; InR for *dpp>InR^act*; Pvr for *dpp>Pvr^act*; Sima for *dpp>Pvr^act+Sima^RNAi*), fold change (log2) of expression level between two genotypes, p-values and significance in RNA-seq analysis are shown.

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
