## [Decision Letter]

Thank you for submitting your article "in vivo Genetic Dissection of Tumor Growth and the Warburg Shift" for consideration by *eLife*. Your article has been reviewed by two peer reviewers, and the evaluation has been overseen by a Reviewing Editor and K VijayRaghavan as the Senior Editor. The following individual involved in review of your submission has agreed to reveal their identity: Leonard I Zon (Reviewer #2).

The reviewers have discussed the reviews with one another and the Reviewing Editor has drafted this decision to help you prepare a revised submission.

Summary:

In this elegant study, the authors observe over-expression of Pvr^act^ (PDGFR/VEGFR) in *Drosophila* wing discs caused tumorous growths and uniquely caused glycolysis induction, measured by LDH enzymatic staining as well as increased LDH transcription via enhancer trap. Importantly, Pvr^act^ expressed in small clones caused cell autonomous LDH induction, allowing the authors to conduct the subsequent genetic dissection in an in vivo system assumed to lack secondary mutations. Using this system, *Drosophila* Hif-α (Sima) is found to be necessary and sufficient to induce LDH expression, although loss of Sima only reduced overgrowth by about 20%. Downstream of Pvr^act^ overexpression, ERK and PI3K signaling were necessary and sufficient to induce increased Sima protein levels and LDH expression. Pvr^act^ mediated glycolytic induction required several downstream factors that regulate translation, leading to the conclusion that activated Pvr, working through PI3K and ERK signaling, leads to increased protein levels of Sima, inducing LDH expression. To assess the transcriptional changes caused by Pvr^act^ over-expression, RNA-sequencing was performed in Pvr^act^ (glycolytic shift), activated InR (no glycolytic shift control), and Pvr^act^ +sima loss of function (glycolytic shift suppressed) wing discs. Transcripts encoding glycolytic enzymes were enriched in Pvr^act^ over-expression wing discs in a Sima-dependent manner. The manuscript then explored the parallel process of oxidative phosphorylation inhibition downstream of Pvr^act^ over-expression via the observed increase in PDHK activity. PI3K and ERK signaling were found to increase PDHK transcript levels, but not PDHK phosphorylation (activation). Src/jun signaling were found to be required for PDHK activation resulting in inactivation of PDH, and inhibition of oxidative phosphorylation. Finally, ROS (created when oxidative phosphorylation is inhibited) reinforced *JNK* signaling and Hif-α protein levels, stabilizing a switch from oxidative phosphorylation to glycolysis downstream of Pvr^act^ over-expression in the wing disc.

A great many of the regulatory steps covered in the manuscript have been described in mammalian models (see review by Pavlova and Thompson, 2016, The Emerging Hallmarks of Cancer Metabolism. Cell Metab 23, 27-47), and some also in *Drosophila* – which are appropriately cited in the manuscript. However, the work reported here has the merit of integrating these in a network to provide a more comprehensive picture of how aerobic glycolysis is brought about.

We recommend publication of this manuscript contingent on the completion of the experiments below, essential for supporting the major conclusions. These experiments, or variations thereof which address our concerns, are eminently do-able within a 2-month period.

Essential revisions:

1) Although it is likely that LDH induction is through Sima in the double PI3K/ERK pathway activation genetic background, in order to conclude LDH induction is via Sima in this context, Sima protein staining and Sima necessity for LDH induction in double pathway activation wing discs should be assessed.

2) Similarly, in order to conclude that LDH induction downstream of Pvr^act^ over-expression relies on translational regulation of Sima, Sima protein staining in wing discs should be assessed in Pvr^act^ over-expression in combination with S6k, lk6/+, and Thor.

In addition, these important points need to be addressed:

An important part of the general scheme that is not totally clear is the postulate that the glycolytic phenotype might be due to mitochondrial dysfunction. This has, indeed, been the accepted view for almost a century, but many recent studies suggest that tumor´s mitochondria are capable of sustaining oxidative phosphorylation. There is no definitive evidence in the manuscript proving that increased lactate levels result from inhibition of oxidative phosphorylation (compared to wild type levels) rather than from a higher glycolytic activity fuelled by increased nutrient uptake (i.e. glucose). Indeed, Ras and PIK3/Akt signalling do regulate glucose uptake. The possibility must be considered that pyruvate conversion to lactate, and the resulting decreased in pyruvate transport into mitochondria, might be a mechanism to avoid an overload of the respiratory chain. To clarify this point glucose uptake and OXPHOS activity should be measured. Without these data it may not be appropriate to refer to a "metabolic shift". The authors are experts in the field and are best placed to see how to address this.

---

## [Author Response]

We recommend publication of this manuscript contingent on the completion of the experiments below, essential for supporting the major conclusions. These experiments, or variations thereof which address our concerns, are eminently do-able within a 2-month period.

Essential revisions:

1) Although it is likely that LDH induction is through Sima in the double PI3K/ERK pathway activation genetic background, in order to conclude LDH induction is via Sima in this context, Sima protein staining and Sima necessity for LDH induction in double pathway activation wing discs should be assessed.

This is a fair point. We had previously demonstrated these results with Pvr^act^ only, with the expectation that the activated PI3K/ERK combination will yield the same, but given that multiple pathways are activated at the same time, the reviewer is correct in asking whether this is indeed the case.

In new experiments, we have now made the proper genetic combinations and show (Figure 3—figure supplement 2) that dual PI3K/ERK activation is indeed sufficient to induce Sima, and also sufficient to regulate LDH expression. (text: subsection “Both ERK and PI3K pathways are necessary for LDH expression”, last paragraph; Figure 3—figure supplement 2).

Briefly, we stained for expression of Sima protein in dpp-Gal4>hRaf^act^+PI3K^act^ wing discs and find that Sima level indeed increases dramatically (Figure 3—figure supplement 2C-D). Sima is also required for LDH induction, since a simultaneous knock down of Sima and co-activation of ERK and PI3K pathways (dpp-Gal4>hRaf^act^+PI3K^act^+Sima^RNAi^) results in a significant reduction in LDH-GFP expression (Figure 3—figure supplement 2H-I).

2) Similarly, in order to conclude that LDH induction downstream of Pvr^act^ over-expression relies on translational regulation of Sima, Sima protein staining in wing discs should be assessed in Pvr^act^ over-expression in combination with S6k, lk6/+, and Thor.

We agree, this issue is now resolved with new data presented in Figure 3—figure supplement 2.

As suggested, we stained for Sima protein in both dpp-Gal4>Pvr^act^+S6K^DN^ and dpp-Gal4>Pvr^act^; lk6^[56]^/+ background and find that Sima accumulation is suppressed in both of these genetic backgrounds (Figure 3—figure supplement 2E-G). For Thor (4EBP), this will require overexpression and not loss of function, and unfortunately the newly acquired chromosomes from Bloomington are not viable as a combination of UAS-Pvr^act^ and UAS-Thor. Nevertheless, we feel that the reviewer’s concern, whether translational control is indeed key to Sima regulation is more directly addressed by loss of function experiments with lk6 and S6K (now shown in Figure 3—figure supplement 2F-G). (Figure: Figure 3—figure supplement 2; Text: subsection “Both ERK and PI3K pathways are necessary for LDH expression”, last paragraph).

In addition, these important points need to be addressed:

An important part of the general scheme that is not totally clear is the postulate that the glycolytic phenotype might be due to mitochondrial dysfunction. This has, indeed, been the accepted view for almost a century, but many recent studies suggest that tumor´s mitochondria are capable of sustaining oxidative phosphorylation. There is no definitive evidence in the manuscript proving that increased lactate levels result from inhibition of oxidative phosphorylation (compared to wild type levels) rather than from a higher glycolytic activity fuelled by increased nutrient uptake (i.e. glucose). Indeed, Ras and PIK3/Akt signalling do regulate glucose uptake. The possibility must be considered that pyruvate conversion to lactate, and the resulting decreased in pyruvate transport into mitochondria, might be a mechanism to avoid an overload of the respiratory chain. To clarify this point glucose uptake and OXPHOS activity should be measured. Without these data it may not be appropriate to refer to a "metabolic shift". The authors are experts in the field and are best placed to see how to address this.

The reviewer is correct. Unfortunately, at this point we can only provide relief to this argument by pointing this out as a caveat of genetic analysis. What the paper shows unambiguously is that PDH activity is lowered and therefore the pyruvate induced TCA cycle must become inefficient. Some cancer cells avoid this situation by bypass mechanisms, such as by up-regulating Gln usage that can result in formation of αKG and citrate.

We do not know if such mechanisms operate in our system, although we do not see any increase in either glucose or glutamine transporters.

We have tried to measure oxygen conversion rate (OCR) and extracellular acidification rate (ECAR) using a Sea Horse XF24-3 system. This technique is optimized for single cell layers and we see we wild variations when we use whole discs for one of many possible reasons. Dissociating the cells will also have uncertain effects on their metabolic status. We are developing methods that work with small quantities of whole tissue, but this will take a long time to complete and is therefore outside the scope of this genetic analysis. We have also, eliminated any nonessential use of the word “shift” to describe the metabolic effect.

In Summary, the text now reflects the following:

1) That the normal pyruvate driven TCA cycle is attenuated (inactive PDH will not allow the pyruvate acetyl CoA conversion).

2) Glycolysis is enhanced (all glycolytic genes are activated, LDH is functional) compared to normal.

3) However, as other mechanisms (chiefly through increased glutamine import) could produce α-ketoglutarate, as in some cancer cells, we could not be sure that ox-phos is indeed down-regulated. We did acknowledge this in our manuscript, but now made clearer in the new version (Discussion, second paragraph).